# Pro-Environmental Behaviors: Determinants and Obstacles among Italian University Students

**DOI:** 10.3390/ijerph18063306

**Published:** 2021-03-23

**Authors:** Annalaura Carducci, Maria Fiore, Antonio Azara, Guglielmo Bonaccorsi, Martina Bortoletto, Giuseppina Caggiano, Andrea Calamusa, Antonella De Donno, Osvalda De Giglio, Marco Dettori, Pamela Di Giovanni, Angela Di Pietro, Alessio Facciolà, Ileana Federigi, Iolanda Grappasonni, Alberto Izzotti, Giovanni Libralato, Chiara Lorini, Maria Teresa Montagna, Liberata Keti Nicolosi, Grazia Paladino, Giacomo Palomba, Fabio Petrelli, Tiziana Schilirò, Stefania Scuri, Francesca Serio, Marina Tesauro, Marco Verani, Marco Vinceti, Federica Violi, Margherita Ferrante

**Affiliations:** 1Department of Biology, University of Pisa, Via S. Zeno 35, 56127 Pisa, Italy; annalaura.carducci@unipi.it (A.C.); andreacalamusa@hotmail.it (A.C.); g.palomba1@studenti.unipi.it (G.P.); marco.verani@unipi.it (M.V.); 2Department of Medical, Surgical Sciences and Advanced Technologies “G. F. Ingrassia”, Catania University, Via Santa Sofia 87, 95123 Catania, Italy; mfiore@unict.it (M.F.); marfer@unict.it (M.F.); 3Department of Medical, Surgical and Experimental Sciences, University of Sassari, Via Padre Manzella 4, 07100 Sassari, Italy; azara@uniss.it (A.A.); madettori@uniss.it (M.D.); 4Department of Health Science, University of Florence, Viale Morgagni 48, 50134 Florence, Italy; guglielmo.bonaccorsi@unifi.it (G.B.); chiara.lorini@unifi.it (C.L.); 5AZIENDA ULSS 6 EUGANEA, Servizio di Prevenzione, Igiene e Sicurezza negli Ambienti di Lavoro (SPISAL), Via Ospedale 22, 35131 Padova, Italy; martina.bortoletto@libero.it; 6Department of Biomedical Sciences and Human Oncology, University of Bari “Aldo Moro”, Piazza G. Cesare 11, 70124 Bari, Italy; giuseppina.caggiano@uniba.it (G.C.); osvalda.degiglio@uniba.it (O.D.G.); mariateresa.montagna@uniba.it (M.T.M.); 7Laboratory of Hygiene, Department of Biological and Environmental Sciences and Technology, University of Salento, 73100 Lecce, Italy; antonella.dedonno@unisalento.it (A.D.D.); francesca.serio@unisalento.it (F.S.); 8Department of Pharmacy, “G. d’Annunzio” University of Chieti-Pescara, Via dei Vestini 31, 66100 Chieti, Italy; pamela.digiovanni@unich.it; 9Department of Biomedical and Dental Sciences and Morphofunctional Imaging, University of Messina, Policlinico Universitario “G. Martino”, Via Consolare Valeria 1, 98100 Messina, Italy; adipietr@unime.it (A.D.P.); afacciola@unime.it (A.F.); 10School of Medicinal and Health Products Sciences, University of Camerino, Via Madonna delle Carceri 9, 62032 Camerino, Italy; iolanda.grappasonni@unicam.it (I.G.); fabio.petrelli@unicam.it (F.P.); stefania.scuri@unicam.it (S.S.); 11Department of Experimental Medicine, School of Medicine, University of Genoa, Via Antonio Pastore 1, 16132 Genoa, Italy; izzotti@unige.it; 12IRCCS Ospedale Policlinico San Martino, Largo Rosanna Benzi 10, 16132 Genova, Italy; 13Department of Biology, University of Naples Federico II, Via Cinthia 21, 80126 Naples, Italy; giovanni.libralato@unina.it; 14Department of Medical, Surgical Sciences and Advanced Technologies “G. F. Ingrassia”, Specialization School of Hygiene and Preventive Medicine, Catania University, Via Santa Sofia 87, 95123 Catania, Italy; ketynicolosi@gmail.com (L.K.N.); graziapaladino2@gmail.com (G.P.); 15Department of Public Health and Pediatrics, University of Torino, Piazza Polonia 94, 10126 Torino, Italy; tiziana.schiliro@unito.it; 16Department of Biomedical, Surgical and Dental Sciences, University of Milan, Via Carlo Pascal 36, 20133 Milan, Italy; marina.tesauro@unimi.it; 17Section of Public Health, Department of Biomedical, Metabolic and Neural Sciences, University of Modena and Reggio Emilia, Via Giuseppe Campi 287, 41125 Modena, Italy; marco.vinceti@unimore.it (M.V.); federica.violi@unimore.it (F.V.); 18Department of Epidemiology, Boston University School of Public Health, Boston, MA 02118, USA

**Keywords:** pro-environmental attitudes, pro-environmental behaviors, environmental health risk perception, functional health literacy, risk communication, internal locus of control

## Abstract

The awareness of citizens concerning the health risks caused by environmental pollution is growing, but studies on determinants of pro-environmental behaviors have rarely examined health-related aspects. In this study, we investigated these determinants using data from a large survey among Italian university students (15 Universities: 4778 filled questionnaires). Besides the health-related aspects, represented by environmental health risk perception and functional health literacy, we considered social and demographic characteristics (gender, area of residence, sources of information, trust in institutional and non-institutional subjects, and students’ capacity of positive actions, indicated as internal locus of control). The attitudes towards pro-environmental behaviors were positive for more than 70% of students and positively related with health risk perception, internal locus of control, and health literacy. The correspondence between the positive attitudes towards pro-environmental behaviors and the real adoption of such behaviors was approximately 20% for most behaviors, except for the separate collection of waste (60%). Such a discrepancy can be attributable to external obstacles (i.e., lack of time, costs, lack of support). The health-related aspects were linked to the pro-environmental attitudes, but to a lesser extent to pro-environmental behaviors, owing to the complexity of their determinants. However, they should be taken in account in planning education interventions.

## 1. Introduction

Today, citizens are involved with environmental pollution in a double role: as victims—that suffer harms from air, water, and food contamination, often with inequalities [1], and as culprits—owing to factors of waste production, traffic, energy consumption, and so on. In fact, the shift of pollution sources from production to consumption processes makes the pro-environmental behaviors of citizens essential for reducing pollution.

The understanding of the determinants underlying the pro-environmental behaviors is a topic that arose of interest since the 1960s, with the aim of increasing them with effective interventions. Different psychological or sociological models have been developed to explain the complexity of such determinants. The theory of planned behavior considers the rational evaluation of consequences as the main determinant of attitudes, intentions, and behaviors, including both hedonic and gain perspectives and taking into consideration the perceived behavioral control. Such a theoretical framework was proposed in 1985 [2], and it is currently used to explain a wide range of pro-environmental behaviors, for example actions for climate change mitigation (i.e., use of public transportation or energy-efficient devices and purchasing of energy-saving appliances) [3,4,5]. More recently, other researchers have proposed various theories and hypotheses of pro-environmental behavior. The value-belief-norm theory has indicated “values” as determinants of both positive attitudes and behaviors, defining such values as “desirable goals that serve as guiding principles in one’s life” [6]. The goal framing theory identified three types of goals: hedonic goals, gain goals, and normative goals [7]. In a metanalysis on phycological determinants of pro-environmental behaviors, Bemberg and Moser [8] explained them as a combination of self-interest, pro-social motives, and moral norms, which are in turn influenced by cognitive, emotional, and social factors. Nevertheless, the studies based on the above-mentioned theoretical models rarely considered factors linked to the human health [6,9,10], although in recent years citizens showed a growing awareness on health risks caused by environmental pollution [11] and this is a very important individual and collective interest that should motivate the environmental protection.

To date, the urgent need of considering health and environment from the “one health” perspective increases the interest in studies on the impact of health risk perception (and its determinants) on pro-environmental attitudes and behaviors. To fill this gap of knowledge the present work was aimed to analyze the impact of health risk perception and functional health literacy on pro-environmental attitudes and behaviors, also taking in account other variables that could influence them, namely socio-demographic characteristics, sources of information, trust in institutional or non-institutional subjects, and internal locus of control. The data for these analyses came from a comprehensive questionnaire survey carried out on university students in 15 Italian cities, whose results on the determinant of risk perception were previously reported in [12]. This further analysis was carried out to test the hypothesis that health risk perception and health literacy were on their turn associated to pro-environmental attitudes and behaviors.

## 2. Materials and Methods

### 2.1. Site Selection and Sampling Techinique

This research was a cross-sectional, nation-wide and multicentric study, with the enrollment of 15 Universities distributed along Italy, in order to cover the different areas in which such nation is traditionally divided (north, center, south, islands), carried out from November 2017 to January 2018. To establish the desirable sample size a software for public service has been used (https://www.epicentro.iss.it/strumenti/SampleSize, accessed on 5 March 2021), considering a target population of 1,530,415 Italian public university students [13], a confident level of 95%, and a margin of error of 1.5% [14]. A sample size of 4257 was considered representative of the target population.

A stratified sampling was employed, based on the University cities (in alphabetic order: Bari, Camerino, Catania, Chieti, Florence, Genoa, Lecce, Messina, Milan, Modena and Reggio Emilia, Naples, Padua, Pisa, Sassari, and Turin) according to a quota sampling of around 300 for each site.

Then, students were enrolled within classrooms or study rooms, until the decided number was reached. The survey instrument was distributed and immediately filled on site by students and collected after compilation in boxes (for anonymity purpose). Overall, 4778 surveys were completed, with a response rate of 99%, attributable to the enrollment strategy. Such number of usable questionnaires was slightly higher than the desirable sample size, giving the great interest shown by the students.

### 2.2. Research Instrument and Data Collection

The study instrument was a self-administered anonymous questionnaire, whose questions were formulated following a deep discussion of the research team and were not previously published, except the question aimed to estimate the functional health literacy [15]. The questionnaire was written in Italian and on average needed 15 min to be filled. It was approved by the Ethical Committee of the University of Milan, then it was set up through a pilot test on 362 students coming from seven Universities among those included in the study, in order to evaluate comprehensibility and acceptability of the questions. The internal consistency of the questionnaire was assessed using Cronbach’s alpha reliability test (Cronbach’s alpha > 0.60 for all the global indexes described below).

The questionnaire was divided into 6 sections and the answers to the first five sections were analyzed in a previous paper [12]. The sixth section of the questionnaire, aimed to study pro-environmental attitudes, behaviors and related obstacles is examined in the present work, also exploring correlations with the other sections. The questionnaire’s sections are described below. The English translation of the complete questionnaire can be found in the Appendix A.

Socio-demographic characteristics: gender, age, place of residence, and the sector of University degree course represented by science–health (biological and environmental sciences, biotechnology, medicine, pharmacy, physics, mathematics, and civil and industrial engineering) and humanistic–legal–social (sociology, political sciences, communication sciences, literature, philosophy, cultural heritage, business economics, economics and finance, and law).Information: sources, trust in them, perceived quality of information, self-evaluation of knowledge on environmental health risks.Environmental health risk perception: estimation of burden of environmental diseases, opinion about the association between environmental factors and some diseases, risk perception towards environmental risks, risk perception towards behavioral risks, general environmental risk perception and self-perception of their own health status, smoking habits.Trust in different subjects: evaluation of the importance of different subjects in pollution reduction and control, evaluation of the real fulfillment of such subjects.Functional health literacy (FHL): measurement of the ability to read and understand information related to health. The understanding of 12 terms was tested by asking participants to place them in the correct section of a stylized body divided into four sections [14].Attitudes and behaviors to reduce and control the environmental pollution and related obstacles. This topic has been explored with five questions, that have been examined in the present study. The five questions and their items, as well as the level of measurement, are described below and summarized in Table 1.

“Level of potential personal support towards environmental interventions” has been investigated through 6 items and the answers were coded according to a Likert 5-point-scale (1 = very low; 2 = low; 3 = neither high nor low; 4 = high; 5 = very high);“Supporting attitude towards measures to reduce the air pollution” has been investigated through 6 items and the answers were coded according to a Likert 4-point-scale (1 = strongly disagree; 2 = disagree; 3 = agree; 4 = strongly agree);“Importance of various citizens behaviors against pollution” (pro-environmental attitudes) has been investigated through 6 items and the answers were coded according to a Likert 5-point-scale (1 = not important; 2 = not very important; 3 = quite important; 4 = very important; 5 = extremely important);“Level of adoption of pro-environmental behaviors” has been investigated through 5 items and the answers were coded according to a Likert 4-point-scale (1 = never; 2 = rarely; 3 = yes, sometimes; 4 = yes, always);“Obstacles against pro-environmental behaviors” have been investigated allowing to choose for each of the above-listed pro-environmental behavior one or more of the following obstacles.

### 2.3. Data Analysis

The answers to the questionnaire were coded as qualitative data or scores, according to the question and analyzed with SPSS 21.0 software (SPSS Inc., Chicago, IL, USA).

The data analysis was articulated in three main parts:

(A) Descriptive results of the answers to the questions of the Section S6, with the total frequencies of answers, according to the Likert scales.

(B) Bivariate analysis to evaluate associations between the answers to the Section S6 and other variables. To this aim, the medians for the Likert scales were calculated. For some questions global indexes were calculated from the sum of scores for the single items (see Appendix A for the items’ list used for the calculation of the global indexes).

From the sixth section of the questionnaire (which represents the focus of the present paper), we calculated the following global indexes:Level of consensus for environmental intervention potentially perceived as negative: global negative attitudes index (GNA);Global support of measures against air pollution (GS);Importance of pro-environmental behaviors, referred as positive attitudes for pro-environmental behaviors (PAPEB);Adoption of pro-environmental behaviors (APEB).

From the section 1 to 5 of the questionnaire (which were previously analyzed [12]), we considered the following variables (global indexes and parameters):Global health risk perception (GHRP) was calculated as global index (Appendix A);Trust in subject actions against pollution, separately for institutional and non-institutional, that are hereafter indicated as trust in action by institutional subjects (TAI) and trust in action by non-institutional subjects (TANI), both calculated as global indexes (Appendix A);Functional health literacy (FHL) score was calculated coding each question as 1 (correct) or 0 (missing or incorrect) and summing the codes of the 12 questions (minimum 0, maximum 12). The total score was divided into two levels: ≤9 (low FHL) and >9 (high FHL) based on its median;Internal locus of control (ILC) was obtained from the question on trust (“How important are the following subjects in protecting the general population from environmental health hazards?”) and the item on citizens importance was considered as an index of the “internal locus of control” based on a Likert 4-point-scale (1 = Not important; 2 = Not very important; 3 = Quite important; 4 = Very important; 5 = Extremely important). Then, ILC was analyzed separately to evaluate its relations with attitudes and behaviors.Internet and social as sources of information (yes/no);Gender (female/male);Area of residence was expressed by grouping the provinces of residence into two main areas: north-center (Camerino, Florence, Genoa, Milan, Modena and Reggio Emilia, Padua, Pisa, Turin) and south islands (Bari, Catania, Chieti, Lecce, Messina, Naples, Sassari).

The bivariate analysis was performed to understand the strength of the association between each of the global indexes calculated in the present work (GNA, GS, PAPEB, APEB) (as well as for the single items used for global indexes’ calculation) and the other variables (global health risk perception index-GHRP, trust in institution index-TAI, functional health literacy-FHL, internal locus of control-ILC, sources of information, gender, and area of residence).

Additionally, any declared obstacles against pro-environmental behaviors were investigated in order to measure their association with the above-mentioned variables. Such analyses were performed using chi-square test, Spearman rank correlation, Student’s t-test or Mann–Whitney U test, as appropriate. Cramer’ V and Spearman’s rho were used to measure the strength of the relationships between the study variables.

(C) A multivariable logistic regression analysis was performed to understand the possible determinants of pro-environmental attitudes (PAPEB) and behaviors (APEB), that have been used as dependent variables, one at time. To this aim, PAPEB and APEB were dichotomized using the median as the cutoff values: low PAPEB = ≤26, high PAPEB = >26; low APEB = ≤15, high APEB = >15. In each model, a total of 10 independent variables were considered, including both global indexes (PAPEB or APEB, GNA, GS, GHRP, FHL, TAI, TANI) and parameters (gender, area of residence, Internet and Social networks as sources of information). Global indexes used as independent variables were in turn dichotomized on the basis of their median values to allow logistic regression analysis (Table 2). In such table, number of students in each category has been calculated. The role of each independent variable on PAPEB or APEB was evaluated in terms of odds ratio (OR) that has been adjusted for all the other variables included in the model (ORadj).

## 3. RESULTS

### 3.1. Descriptive and Univariate Analysis for Attitudes

The answers to the questions exploring attitudes towards environmental initiatives showed a general disagreement (GNA index = 13 ± 5) towards actions aimed at introducing structures perceived as potentially impacting on health, such as a new landfill, an oil/gas pipeline, a new highway, a new incinerator, and a new high voltage line close to their neighborhood (Figure 1).

People with a lower trust in the real fulfillment of actions for environmental protection showed higher negative attitudes, globally and for single items. No differences were found according to gender, area of residence, FHL, global health risk perception index, internal locus of control, and sources of information (Appendix A).

On the contrary, the agreement for the listed pro-environmental initiatives was generally high: a natural park was supported by more than 80% of students, mainly by students with high FHL and with a higher internal locus of control.

Similarly, the measures to limit air pollution were supported by more than 60% (Figure 2), except for the introduction of toll parking (less than 30%). All the other initiatives were more supported by students with higher FHL and higher internal locus of control (Appendix A). Limitation of vehicular traffic in the city and closure of city center were more supported by people with a higher level of risk perception. No differences were found on the basis of gender, FHL, or area of residence. The index of global support towards measures to limit air pollution was positively associated with the FHL, the global health risk perception index, the global trust in institutions, and the internal locus of control.

Overall, the attitudes towards pro-environmental behaviors were very positive (Figure 3): more than 70% think that they are extremely against pollution. The positive attitudes towards all the listed behaviors were higher for students with higher health risk perception index and higher internal locus of control (Appendix A). The separate collection of waste was also considered more important by people more trusted in institutions. The global index of positive attitudes for pro-environmental behaviors was positively associated with female gender, FHL, health risk perception index, trust in institutions fulfilment, and internal locus of control.

### 3.2. Descriptive and Univariate Analysis for Pro-Environmental Behaviors and Obstacles

The adoption of frequent pro-environmental behaviors was globally lower than the positive attitudes towards them (Figure 4). Only the separate collection of waste is adopted very frequently by more than 60%, according to the positive attitudes towards it and without differences in relation to gender, area of residence, FHL, global health risk perception index, and trust in institution indexes (Appendix A). The use of less polluting fuels, the reduction of energy consumption and the choice of low environmental impact products were more frequent for people with higher health risk perception index. The global index for the adoption of pro-environmental behaviors was higher for people with higher health risk perception index and slightly influenced by internal locus of control.

Among the obstacles against the adoption of pro-environmental behaviors, the lack of support from institutions was the most cited, except for the purchase of products with low environmental impact, that is mostly hampered by costs. These two previous mentioned obstacles also reduce the use of less polluting fuels (Table 3).

The associations between the perceived obstacles for behaviors and other variables (gender, area of residence, functional health literacy, health risk perception, trust in institution fulfillment, internal locus of control, internet and social as sources of information) were quite weak (Appendix A).

The most evident influences came from the area of residence: people living in the south islands declared more frequently the lack of time for separate collection of waste and all the considered obstacles for the energy consumption. Instead, the lack of support from institutions was more frequently declared by students living in north centre. Students living in the south islands seemed to be less concerned about the lack of support from institutions also for the use of less polluting fuels and for buying low impact products, while they declared more frequently the lack of support from familiars and friends.

Some differences were found also between the internet and social users and non-users regarding the use of less polluting fuels and buying low impact products: the internet and social users seemed more influenced by the costs and less by the lack of time and the lack of support by institutions.

### 3.3. Multivariate Analysis for Attitudes and Behaviors

The multiple logistic regression analyses of positive attitudes towards pro-environmental behaviors (PAPEB) index (dependent variable) showed that this index was lower for people with lower global health risk perception index, lower trust in action both by institutional and non-institutional subjects (TAI and TANI indexes), lower FHL, lower adoption of pro-environmental behaviors (APEB index), and lower global support (GS) index. On the contrary, PAPEB was higher for people with lower GNA, which represents potentially negative attitudes towards environmental initiatives (see Table 4 and Appendix A for detailed results).

On the other hand, the multiple logistic regression analysis of adoption of pro-environmental behaviors (APEB) index (dependent variable) showed that it was lower for people with lower trust in action by non-institutional subjects (TANI), lower global support for positive actions (GS), lower positive attitudes for pro-environmental behaviors (PAPEB), principal sources of information different from Internet and social, and living in south islands. On the contrary, it was higher for people with lower GNA, as in the case of PAPEB (Table 4 and Appendix A for detailed results).

## 4. Discussion

In recent years young people, mainly students, became increasingly sensitive about the environmental issues, as demonstrated by movements such as Friday for Future or School Strikes for Climate [11], but the effective promotion of pro-environmental behaviors has not been proven. Some studies have indicated that the age group 18–24 years is less concerned about environment and less inclined to adopt pro-environmental behaviors, even when concerned [16]. University students are often surveyed about these issues because they are considered the future decision makers, but also, they are very accessible for administering questionnaires [17]. In our survey, on the whole, a high proportion of students showed very positive attitudes towards pro-environmental initiatives and actions, while they mostly were opposed to potentially negative interventions. Similar results were reported by other studies in analogous populations, even in different countries [18].

On the other hand, except for the separate waste collection, the frequency of people adopting positive behaviors was about half of the one of people declaring their importance. This finding was sometimes confirmed by the literature, while in other cases environmental behaviors were aligned to attitudes at the various degrees of correspondence [19,20]. The reasons of this contradiction could be found in the complexity of determinants for pro-environmental behaviors, that different psychological or sociological models have tried to explain [2,6,7,8].

As already said, the studies about determinants of pro-environmental attitudes and behaviors rarely have taken in account the health-related aspects, even if today the deep relations between environment and health are more and more evident. In the present work we have focused on two health-related aspects, the environmental health risk perception and the FHL and the results are discussed below.

### 4.1. Health Risk Perception

The risk perception can be considered a motivation to act either in an egoistic perspective or from a social values point of view, depending on the faced risks. The importance of risk perception in influencing pro-environmental attitudes and behaviors has been demonstrated by many studies in different populations, times, and countries [21]. These studies considered general environmental risks (including both the ecological and the health-related ones), whereas our investigation focused on health-related risk perception deriving from environmental threats: the impact of this factor was evident both on attitudes and behaviors, although more evident on the first ones. This last relation remained clear even in the multivariate analysis, considering other possibly influencing variables, while it became less evident for behaviors.

As regards the single behaviors, people with higher health risk perception adopted more frequently the use of less polluting fuels, the reduction of energy consumption, and the choice of low environmental impact products. This result agrees with other studies, although with variations in the strength of association: some authors showed that environmental concerns were linked to more frequent positive behaviors as a whole, while this association was weak for waste, green products, and energy consumption [6].

### 4.2. Functional Health Literacy and Sources of Information

In the present study, the ability in understanding and using health-related information (FHL) appeared positively related to attitudes, even if not clearly on behaviors. Even if studies on the environmental awareness, till now, have rarely considered the FHL, it is now becoming increasingly important, and the new definition of “environmental health literacy” has been coined [22], including not only the functional dimension, but also the critical and interactive ones, as well as other social and public health factors.

For instance, environmental health literacy was related to preventive behaviors among people exposed to environmental risks [23].

However, many studies exploring determinants of pro-environmental attitudes and behaviors have demonstrated the importance of knowledge [6,16], showing that it contributes to the formation and activation of moral norms and represents a fundamental precondition of them [3]. On the other hand, most researchers found that environmental knowledge and environmental awareness can be associated only to a small fraction of pro-environmental behaviors, because the information alone is not sufficient to bring about action and change [24].

Another important issue about the awareness is the source of information: besides formal education, some other sources can increase awareness and induce positive attitudes and, sometimes, behaviors. In our survey the most frequent source of information was Internet and social networks, whose use was slightly associated with the adoption of positive behaviors.

### 4.3. Other Variables Influencing Pro-Environmental Attitudes and Behaviors

In the present work, the importance attributed to pro-environmental behaviors (represented by PAPEB) was associated not only with environmental health risk perception and functional health literacy, but also with trust in institutional and non-institutional subjects and, slightly, with female gender. On the other hand, pro-environmental behaviors (APEB) were influenced not only by positive attitudes and, slightly, by environmental health risk perception and health literacy, but APEB was also positively associated with trust in action by non-institutional subjects, global support for positive actions, source of information (internet and social networks), and area of residence (south islands).

The role of age, gender, income, level of education, political tendency, and area of residence in determining pro-environmental attitudes and behaviors has been demonstrated in different populations [6,25,26].

Moreover, we found that people attributing importance to citizens actions (assumed to indicate a high internal locus of control) had more positive attitudes and (slightly) more positive behaviors, confirming findings of other researchers [6,8,27]. Nevertheless, the self-confidence in the efficacy of personal behaviors can vary according to the age of the target population, the geographical area, and time, as observed in various surveys [12,28,29]. These variations may be related to different factors such as political, social, and economic constraints. Cultural and social differences through the countries, in fact, could justify the influence of the area of residence on attitudes and behaviors, as also showed by our findings.

Moreover, we observed that the trust in institutional and non-institutional subjects is an important incentive to positive attitudes and behaviors, in agreement with the observation that if pro-environmental values are endorsed by the society, this can promote positive actions, as well as attitudes [11].

### 4.4. Single Pro-Environmental Behaviors and Obstacles against Them

Even if pro-environmental behaviors have been often considered as a whole, each of them can have different determinants. For example, we have found that the separate waste collection is the most frequently adopted behavior, while the habit to buy environmentally friendly products is the least ones. Poškus [30] supposed that the compliance to different environmental behaviors was influenced by the self-interest: conserving water and electricity might be enhanced by the possibility of saving money, while recycling and using more sustainable transportation would requires more efforts.

Besides the overall influence of the risk perception, the knowledge and other socio-psychological factors, behaviors are influenced by the barriers that people encounter, classified by Kollmuss and Agyeman [31] into two groups: the internal and the external ones. Among the external barriers, a great importance is attributed to the lack of support from institutions in terms of infrastructures (i.e., public transportations, efficient waste collection and disposal) and of incentives for pro-environmental behaviors and advertisements. Accordingly, in our study, the lack of support from institutions was the most cited obstacle towards the investigated behaviors, except for buying low impact products, for which cost was the most important barrier. This result on obstacles coming from university students agrees with the one previously found in general population [28].

On the other hand, the economic obstacles can be important, but they are often also linked with social, infrastructural, and psychological factors [6]. The analysis of factors influencing obstacles shows the great complexity of this framework and indicates the need for specific investigations when pro-environmental programs are designed: area of residence and sources of information seem to be the most important in our study.

### 4.5. Limitation of the Study

Our study analyzed the influences of health risk perception and health literacy on pro-environmental attitudes and behaviors, without applying any of the existing psychological and sociological models to explaining them. Therefore, our data should be confirmed by a further research, designing a more complex framework to include more determinants. The risk perception is highly dependent on contingent factors, variable in time and space, then some of our findings could be changed in the present situation, especially considering the pandemic and its general impact on the health risk perception. Similar surveys should be repeated at intervals to evidence the time evolution of the data. Another limitation of the study is the use of a simple test for the functional health literacy. This test is quite sensitive for risk perception and positive environmental attitudes, but it could be unable to point out other important determinants of behaviors, that would be better highlighted by a more complex index of environmental health literacy.

## 5. Conclusions

In the present work, the importance attributed to pro-environmental behaviors (i.e., index of global positive attitudes towards environment-PAPEB) was associated with both environmental health risk perception and functional health literacy, besides other variables. Nevertheless, the same associations were weaker for behaviors (APEB) confirming the well-known assumption that their adoption depends not only on positive attitudes, but also on other variables including internal and external factors [31]. Such results cannot be fully compared with other similar investigations because health-related aspects have not been specifically addressed before. Nevertheless, some results agree with other environmental surveys, such as the correlation between positive attitudes and female gender, trust in institutions fulfilment, and internal locus of control.

At present, the awareness of citizens on health risks caused by environmental pollution is growing and worldwide institutions are affirming the importance of inter-relations between health and environment, promoting the wider perspective of “one health” and “planetary health”, confluent in the “agenda 2030 for sustainable development”.

Then, during the development of interventions to promote pro-environmental behaviors, the target population should be studied also including the health-related aspects, with the approach of social marketing, that has been successful in overcoming the gap between attitudes and behaviors in sustainability projects [31].

## Figures and Tables

**Figure 1 ijerph-18-03306-f001:**
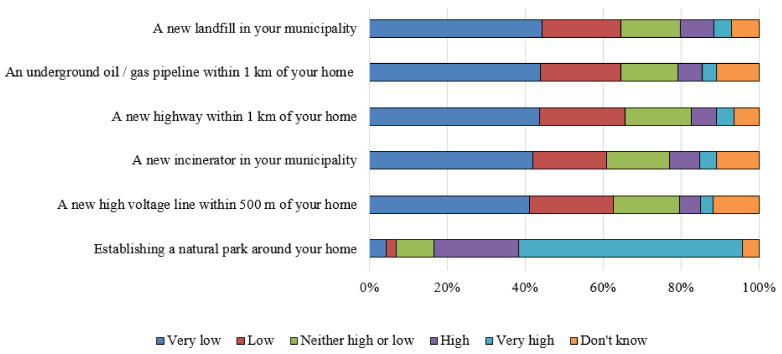
Level of potential support for environmental initiatives.

**Figure 2 ijerph-18-03306-f002:**
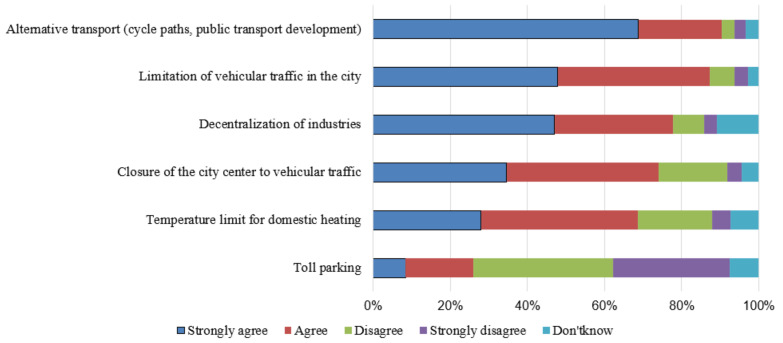
Level of support of measures to limit air pollution.

**Figure 3 ijerph-18-03306-f003:**
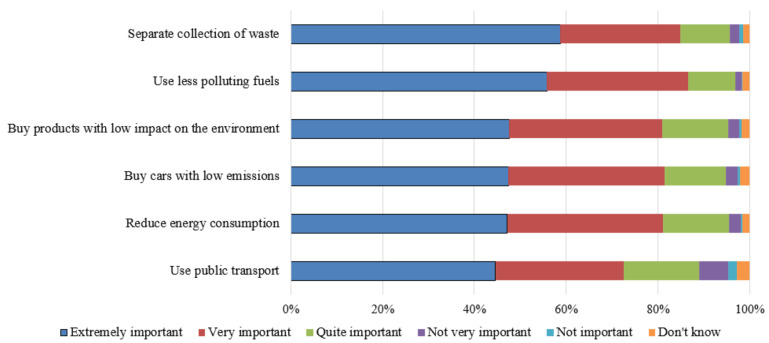
Level of importance attributed to behaviors of citizens in the fight against pollution.

**Figure 4 ijerph-18-03306-f004:**
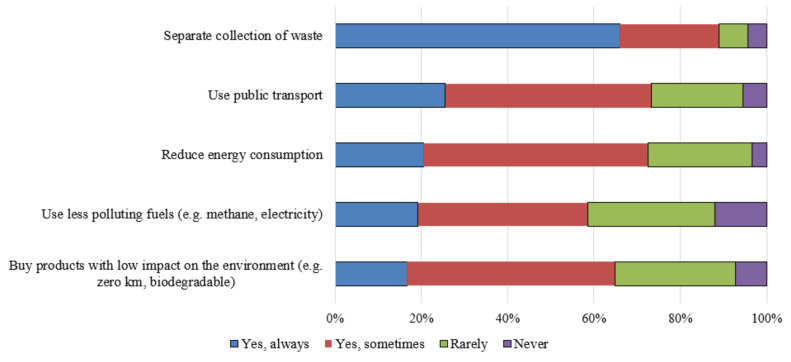
Frequency of the adoption of pro-environmental behaviors.

**Table 1 ijerph-18-03306-t001:** Questions, items, and level of measurement used to explore attitudes and behaviors to reduce and control the environmental pollution.

Topic	Question	Items	Answer Coding
Level of potential personal support towards environmental interventions	Indicate your level of potential support for the following initiatives	− A new incinerator in your Municipality; − A new landfill in your Municipality; − A new high voltage line within 500 m from your home; − An underground oil/gas pipeline within 1 km of your home; − A new highway within 1 km of your home; − Establishing a natural park around your home	Likert 5-point-scale where ‘1’ indicates very low support and ‘5’ very high support
Supporting attitude towards measures to reduce the air pollution	To what extent do you support the following measures to limit air pollution?	− Limitation of vehicular traffic in the city; − Closure of the center to vehicular traffic; − Toll parking; − Alternative transport (cycle paths, public transport development); − Temperature limit for domestic heating − Decentralization of industries	Likert 4-point-scale where ‘1’ indicates strong disagreement and ‘4’ strong agreement.
Importance of various citizens behaviors against pollution	In your opinion, how important are the following behaviors of citizens in the fight against pollution?	− Separate collection waste; − Use fewer polluting fuels; − Buy products with low impact on the environment; − Reduce energy consumption; − Buy cars with low emission; − Use public transport	Likert 5-point-scale where ‘1’ indicates no importance and ‘5’ extremely importance
Level of adoption of pro-environmental behaviors	How often have you adopted the following behaviors?	− Separate collection waste; − Use public transport; − Reduce energy consumption; − Use fewer polluting fuels (i.e., methane, electricity); − Buy products with low impact on the environment (i.e., zero km, biodegradable)	Likert 4-point-scale where ‘1’ indicates that the behavior is never adopted and ‘4’ always adopted
Obstacles against pro-environmental behaviors	What obstacles do you find in implementing them? (report obstacles, even more than one, for each behavior)	− Separate collection waste; − Use public transport; − Reduce energy consumption; − Use fewer polluting fuels (i.e., methane, electricity); − Buy products with low impact on the environment (i.e., zero km, biodegradable)	Choose one or more of the following obstacles: Lack of support from institutions; Lack of support from family/neighbors/acquaintances; Lack of time; Mistrust in effectiveness; Costs

**Table 2 ijerph-18-03306-t002:** Global indexes and parameters used as independent variables in the multivariable logistic regression models of pro-environmental attitudes (PAPEB) and behaviors (APEB). The percentage values refer to the total study population (4778 students).

Global Indexes and Parameters	Dichotomization	N° 4778 (%)
Global negative attitudes (GNA)	High (> 12)	990 (21.3%)
	Low (≤ 12)	3660 (78.7%)
Global support (GS)	High (> 18)	1954 (40.9%)
	Low (≤ 18)	2824 (59.1%)
Gender	Female	3106 (65%)
	Male	1672 (35%)
Area of residence	North-center	2055 (43%)
	South islands	2723 (57%)
Internet and Social networks as sources of information	Yes	3713 (77.7%)
	No	1065 (22.3%)
Global Health Risk Perception (GHRP)	High (> 75)	4179 (87.8%)
	Low (≤ 75)	581 (12.2%)
Functional Health Literacy (FHL)	High (> 9)	2102 (44%)
	Low (≤ 9)	2676 (56%)
Trust in action by institutional subjects (TAI)	High (> 21)	1474 (30.8%)
	Low (≤ 21)	3304 (69.2%)
Trust in action by non-institutional subjects (TANI)	High (> 15)	895 (18.7%)
	Low (≤ 15)	3883 (81.3%)

**Table 3 ijerph-18-03306-t003:** Perceived obstacles against pro-environmental behaviors (percentages of total respondents are reported for each statement. More than one obstacle may have been reported).

Obstacles	Lack of Support from Institutions (%)	Lack of Support from Family/Neighbors/Acquaintances (%)	Lack of Time (%)	Mistrust in Effectiveness (%)	Costs (%)
**Behaviors**					
Separate collection of waste	46.1	19.4	13.4	18.6	2.4
Use public transport	46.6	3.7	16.4	16.7	16.5
Reduce energy consumption	40.6	26.7	11.8	9.6	11.3
Use less polluting fuels (e.g., methane, electricity)	41.4	13.5	7.5	7.4	30.2
Buy products with low impact on the environment	21.0	13.5	7.8	9.4	48.3

**Table 4 ijerph-18-03306-t004:** Results of the logistic regression analyses expressed as adjusted odds ratio (ORadj) with 95% confidence interval (95%CI) for low PAPEB (<26) and low APEB (<15) according to independent variables. The ORs represent the risk of lower PAPEB (or lower APEB) in comparison with the reference categories (indicated by asterisks) of independent variables.

Independent Variables (in Parentheses the Reference Category)	Risk of Lower PAPEB ORadj (95% CI) *	Risk of Lower APEB ORadj (95% CI) *
Positive attitudes toward behaviors—PAPEB (High)	NA	2.44 (2.14–2.78)
Global adoption of behaviors—APEB (High)	2.44 (2.14–2.78)	NA
Global negative attitudes—GNA (High)	0.76 (0.67–0.87)	0.83 (0.73–0.94)
Global support—GS (High)	2.78 (2.44–3.16)	1.31 (1.15–1.49)
Gender (Female)	1.07 (0.95–1.24)	0.96 (0.84–1.09)
Area of residence (north-center)	1.00 (0.88–1.13)	1.14 (1.01–1.30)
Internet and social as sources of information (No)	0.98 (0.84–1.14)	1.23 (1.06–1.42)
Global health risk perception—GHRP (High)	2.48 (1.99–3.09)	1.18 (0.96–1.45)
Functional health literacy—FHL (High)	1.20 (1.06–1.37)	1.06 (0.94–1.21)
Trust in action by institution—TAI (High)	1.19 (1.03–1.38)	1.06 (0.91–1.22)
Trust in action by non-institutional subject—TANI (High)	1.51 (1.27–1.80)	1.45 (1.22–1.72)

* Each odds ratio is adjusted for all other variables inserted in the model and showed in the table. NA stands for “not applicable”, since PAPEB and APEB has been used in turn as dependent variable.

## Data Availability

Data are contained within the article or Appendix A.

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
