# Peer review of "Pro-Environmental Behaviors: Determinants and Obstacles among Italian University Students"

_ijerph, 2021, doi:10.3390/ijerph18063306_

Round 1

Reviewer 1 Report

  1. Authors write at the very beginning of the paper that it is based on the pro-environmental Theory of Planned Behavior, but then no comprehensive review of the relevant literature on the topic can be found. A theoretical framing paragraph on that extensive literature would be appreciated. At the moment, only a reference to Ajzen can be found at [2]
  2. At page 3&4 authors list a series of items that were investigated in the empirical analysis. It is not clear though where those items come from: were they formulated by the authors? Or were they taken from previous literature? It would be helpful anyway to turn the list of items into a table in which the source of each item that was used is made explicit. 
  3. Authors often refer to previous research [8] but I strongly think that the paper should stand on its own. It would be appreciated that at least the essential steps of that research were recalled and discussed where necessary.
  4. The role of attitude in the empirical analysis is not totally clear. In the Theory of Planned Behavior it is a predictor of one’s pro-environmental behavior (see for example Brody, et al., 2016; Chen, 2016; Masud et al., 2016). In this paper it is used both as a dependent and as an explanatory variable. Is this the same measure of attitude? And if not, what are the differences between the one used as explanatory and the dependent one?
  5. Tables should be self-explaining
  6. For ease of reading, the univariate results should be clearly separated from the multivariate analysis. Authors should therefore divide section 3 into descriptive statistics and results of the multivariate analysis.

Suggested additional references:

Brody, S., et al. (2012). Examining the willingness of Americans to alter behaviour to mitigate climate change, Climate Policy, 12(1), 1-22. Doi: 10.1080/14693062.2011.579261

Chen, M., (2016). Extending the theory of planned behavior model to explain people's energy savings and carbon reduction behavioral intentions to mitigate climate change in Taiwanemoral obligation matters, Journal of Cleaner Production, 112, 1746-1753. Doi: 10.1016/j.jclepro.2015.07.043 

Masud, M.M., et al. (2016), Climate change issue and theory of planned behaviour: relationship by empirical evidence. Journal of Cleaner Production, 113, 613-623. Doi: /10.1016/j.jclepro.2015.11.080

Reviewer 2 Report

  1. The theoretical foundations can and should be expanded in a way that leads to the exposition of the elements that give rise to the foundation of the research.
  2. The methodological approach described has many gaps:
    1. When describing the methodology, the formal scientific procedure for its description and presentation has not been followed.
    2. The methodology that is applied in the research process is not expressed.
    3. Lack of research objectives,
    4. There are no hypotheses.
    5. There are no data to support the sampling.
    6. The procedure followed is not stated.
    7. The instruments are not based on scientific data for reliability and validity.
    8. There is no adequate expression on the research design.
    9. The results are a data set that is not efficiently articulated, because there is no design that expresses well the logical order of the results.

The gaps expressed leave the discussion with problems of interpretation and credibility.

Conclusions should be raised prior to discussion.

All of which makes me advise against publishing this article. For this, it would have to be rewritten again

Round 2

Reviewer 2 Report

I have gone over to review again, as I promised, the manuscript ID: ijerph-1123281 "Pro-environmental behaviors: determinants and obstacles among Italian university students"

For this reason and once it has been revised again, come to observe that the suggestions made in the revision of the first text have been taken into account and that they have been answered with amplitude and generosity.